# Heating a dipolar quantum fluid into a solid

J. Sánchez-Baena [1,2] ✉, C. Politi [3,4], F. Maucher [1,5], F. Ferlaino [3,4] &
T. Pohl [1] ✉

Raising the temperature of a material enhances the thermal motion of particles. Such an increase in thermal energy commonly leads to the melting of a solid into a fluid and eventually vaporises the liquid into a gaseous phase of matter. Here, we study the finite-temperature physics of dipolar quantum fluids and find surprising deviations from this general phenomenology. In particular, we describe how heating a dipolar superfluid from near-zero temperatures can induce a phase transition to a supersolid state with a broken translational symmetry. We discuss the observation of this effect in experiments on ultracold dysprosium atoms, which opens the door for exploring the unusual thermodynamics of dipolar quantum fluids.

A supersolid is an exotic phase of matter in which particles develop regular spatial order and simultaneously support the frictionless flow of a superfluid. Having evaded experimental verification for several decades[1], supersolidity can now be observed in Bose−Einstein condensates of ultracold atoms with finite-range interactions[2−6]. Spontaneous symmetry breaking in these systems occurs in the form of regular periodic patterns of the condensate density as first predicted by Gross in 1957[7]. One would thus expect the lowest possible temperatures to provide optimal conditions for supersolidity by ensuring a high degree of phase coherence and maximal population of the Bose−Einstein condensate. On the contrary, we demonstrate here that thermal fluctuations in dipolar condensates do not merely diminish global phase coherence but can instead facilitate the formation of periodic modulations of the condensate density. This finding sheds light on recent experimental observations[8] and reveals an unusual fluid-solid phase transition, whereby a supersolid state of matter emerges upon increasing the temperature.

As we shall see below, this surprising behavior arises from the anisotropic nature of the dipole−dipole interaction

$$V_{dd}(\mathbf{r}) = \frac{C_3}{4\pi} \frac{1 - 3\cos^2\theta}{r^3},$$ (1)

which has repulsive as well as attractive contributions, depending on the angle $\theta$ between the dipolar orientation and the distance vector $\mathbf{r}$ of the two atoms. The interaction strength $C_3$ and the atomic mass $m$ define a length scale $a_d = mC_3/(12\pi\hbar^2)$ that competes with the

scattering length $a$ of the short-range interaction between the atoms. This competition between $a_d$ and $a > 0$ can cause the condensate to collapse when the stabilizing short-range repulsion is not sufficient to overcome the attractive part of the dipole−dipole interaction between the atoms[9−11]. Subsequent experiments[12−14] have however found a higher level of stability, which arises from quantum fluctuations[15,16] that prevent the otherwise inevitable collapse of the condensate[17−19]. In fact, the balance of attraction and repulsion effectively enhances the role of quantum fluctuations[18] beyond the semiclassical mean-field physics of weakly interacting quantum gases. This yields a unique setting that has revealed rich physics and a host of new quantum states, from self-bound quantum droplets[13,14,20] and supersolid phases[3−5,21−23] to complex patterns in two-dimensional fluids[24,25].

Given this striking role of quantum fluctuations in dipolar Bose−Einstein condensates, one may also anticipate significant effects of thermal fluctuations despite the ultralow temperatures that are required to reach quantum degeneracy. Here, we demonstrate that this is indeed the case and show how increasing thermal fluctuations can drive an unusual phase transition to a supersolid state.

## Results and discussion

### Thermodynamics of dipolar Bose−Einstein condensates
Let us first consider the grand canonical potential $\Omega$ of the system at a finite temperature $T$. For a weakly interacting gas with a high fraction of atoms in the condensate, one can use Bogoliubov theory to determine $\Omega$. This yields simple expressions for infinitely extended homogeneous systems[26] that can be applied to describe trapped

[1]Center for Complex Quantum Systems, Department of Physics and Astronomy, Aarhus University, DK-8000 Aarhus C, Denmark. [2]Departament de Física, Universitat Politècnica de Catalunya, Campus Nord B4-B5, 08034 Barcelona, Spain. [3]Institut für Quantenoptik und Quanteninformation, Österreichische Akademie der Wissenschaften, Innsbruck, Austria. [4]Institut für Experimentalphysik, Universität Innsbruck, Innsbruck, Austria. [5]Departament de Física, Universitat de les Illes Balears & IAC-3, Campus UIB, E-07122 Palma de Mallorca, Spain. ✉ e-mail: jsbaena@phys.au.dk; pohl@phys.au.dk

inhomogeneous gases within a local density approximation. Hereby, one determines the Bogoliubov excitation spectrum and all relevant observables for a homogeneous particle density $\rho$, which is then identified as $\rho \equiv |\psi(\mathbf{r})|^2$ with the local condensate wave function $\psi(\mathbf{r})$ at a given position $\mathbf{r}$. This permits to express the grand canonical potential as

$$\Omega = E_0 + \frac{k_B T}{(2\pi)^3} \int d\mathbf{r} \int d\mathbf{k} \ln\left(1 - e^{-\frac{\varepsilon_\mathbf{k}(\mathbf{r})}{k_B T}}\right), \qquad (2)$$

where $k_B$ denotes the Boltzmann constant and $E_0$ is the zero-temperature grand canonical energy that contains the mean-field interaction energy and leading order corrections due to quantum fluctuations[15], i.e. small occupations of excited states above the formed Bose–Einstein condensate. The dispersion $\varepsilon_\mathbf{k} = \sqrt{\tau_\mathbf{k}(\tau_\mathbf{k} + 2|\psi(\mathbf{r})|^2 \tilde{V}(\mathbf{k}))}$ of these excitations is determined by the kinetic energy $\tau_\mathbf{k} = \hbar^2 k^2/(2m)$ of the atoms and the Fourier transform $\tilde{V}(\mathbf{k}) = \frac{4\pi\hbar^2 a}{m} + \tilde{V}_{dd}(\mathbf{k})$ of their total interaction potential.

Minimizing $\Omega$ with respect to $\psi(\mathbf{r})$ then yields a nonlinear wave equation that accounts for quantum as well as thermal fluctuations (see "Methods" section). At zero temperature, it describes the mean-field physics of the condensate and captures leading-order effects of quantum fluctuations through an effective density-dependent potential $H_{qu}$[18] that increases the energy of the system. The second term in Eq. (2) yields an additional potential

$$H_{th}(\mathbf{r}) = \int \frac{d\mathbf{k}}{(2\pi)^3} \tilde{V}(\mathbf{k}) f_\mathbf{k}(\mathbf{r}) \frac{\tau_\mathbf{k}}{\varepsilon_\mathbf{k}(\mathbf{r})}, \qquad (3)$$

that accounts for finite-temperature effects. It describes the interaction between the condensate and thermally created excitations that populate Bogoliubov modes according to the Bose distribution $f_\mathbf{k} = 1/(e^{\varepsilon_\mathbf{k}/k_B T} - 1)$. The resulting form of the finite-temperature extended Gross–Pitaevskii equation (TeGPE) agrees with the result of Hartree–Fock Bogoliubov theory[26,27], and includes relevant fluctuation terms that are commonly neglected within the Popov approximation[28] (see the "Methods" section).

## Temperature effects in the thermodynamic limit

We can now use this framework to study an elongated atomic gas that is confined harmonically in the $x-y$ plane and extends infinitely in the $z$-direction without confinement along the $z$-axis. Figure 1a shows the thermodynamic phase diagram obtained by simulating the imaginary time evolution of the TeGPE at a fixed chemical potential $\mu$ (see "Methods" section). At zero temperature, we find a superfluid-supersolid quantum phase transition, with a co-existence region that is expected for a first-order phase transition[29]. While increasing the temperature may generally be expected to melt the supersolid phase[30], we find instead that it shifts the transition towards weaker dipole–dipole interactions. As a result, heating the system effectively drives a phase transition from a fluid into a solid phase.

We can understand this effect from the excitation spectrum of the condensate in the superfluid phase. To this end, we solve the time-dependent TeGPE within linear response theory to find the excitation spectrum $\omega_{k_z}$ for periodic plane-wave excitations along the $z$-direction. As shown in Fig. 2, the obtained dispersion exhibits the expected roton-maxon form[31–35], known from low-temperature helium[36] and Bose-Einstein condensates with finite-range interactions[37–40]. The local minimum at finite momenta supports the formation of roton quasi-particles, which were introduced by Landau as elementary vortices to describe superfluidity in $^4$He[36]. Experiments show that the roton minimum in helium decreases with increasing temperature[41] due to roton-roton scattering[42]. Yet, the roton energy remains sizable at the transition to a normal-fluid phase[41], beyond which it only varies weakly with temperature. The presence of a Bose–Einstein condensate in

dilute dipolar superfluids, however, enhances the effect of thermal fluctuations due to the larger energy scale of the interaction between Bogoliubov excitations and the condensate. A similar effect is found for atoms with light-induced interactions and predicted to lower the roton minimum and cause enhanced condensate depletion[43]. In the present case, we find a thermal softening of the roton mode that can drive an instability of the superfluid and thereby cause the formation of a supersolid phase with increasing temperature. Hereby, the depicted lowering of the roton minimum renders density modulations energetically more favorable as the temperature increases, which eventually triggers a transition to a periodically modulated phase.

We can gain further intuition about the underlying mechanism by closer inspection of the two fluctuation energies $H_{qu}$ and $H_{th}$ that both contribute a local nonlinearity to the wave equation for $\psi(\mathbf{r})$. $H_{qu} > 0$ is the Lee–Huang–Yang correction to the equation of state[15,16], and raises the ground state energy due to the small condensate depletion caused by the atomic interactions. It therefore increases for higher particle densities and stronger interactions, as shown in Fig. 2b. Consequently, $H_{qu}$ generates an effective repulsion that stabilizes the condensate against collapse[18], and shifts the roton instability towards higher densities and stronger dipole-dipole interactions. On the contrary, $H_{th}$ increases as we lower the density of the condensate [see Fig. 2b]. This behavior is readily understood as follows.

Decreasing the condensate density increases the fraction of thermally excited, non-condensed atoms[27]. In the limit where this fraction remains small, such an increase implies a larger potential energy due to interactions with the thermal atoms. It therefore contributes a positive energy correction that decreases upon increasing the density $\rho = |\psi|^2$ of the condensate. As a result, thermal fluctuations energetically favor higher condensate densities, such that $H_{th}$ has a focusing effect on the condensate wave function which lowers the

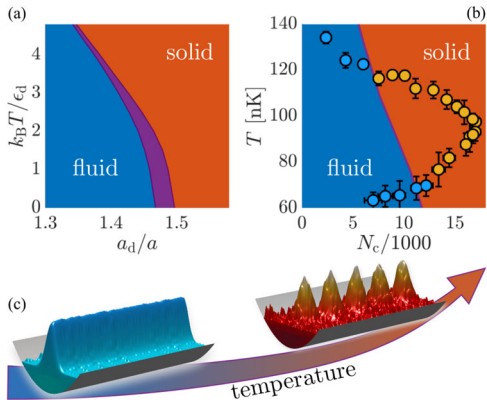

**Fig. 1 | Temperature-driven formation of supersolidity.** Heating a dipolar quantum fluid can lead to the emergence of a supersolid phase of matter, as illustrated schematically in **c**. **a** This is demonstrated in the thermodynamic phase diagram for an infinitely elongated Bose–Einstein condensate in a radial harmonic trap without axial confinement. In between the superfluid (blue) and supersolid (red) region, both phases coexist (purple region) as characteristic for a first-order phase transition. The calculations were performed for a fixed chemical potential $\mu/\epsilon_d = 1$, where $\epsilon_d = \hbar^2/((12\pi)^2 ma_d^2)$ parametrizes the characteristic energy scale of the dipole-dipole interactions. Measurements of supersolid formation during the cooling of a gas of dysprosium atoms[8] indicate the temperature-driven emergence of supersolidity. The color of the dots in **b** indicates the measured density modulation $\mathcal{M}$ whereby orange dots correspond to a large contrast $\mathcal{M} > 0.2$ and blue corresponds to an unmodulated condensate with $\mathcal{M} < 0.2$. Our observations agree well with the theoretically predicted transition (purple line) and show that supersolidity indeed arises with increasing temperature while keeping the number, $N_c$, of condensed atoms as well as the interaction strengths ($a_d/a = 1.46$ and $a = 4.7$ nm) fixed. The error bars in **b** indicate the statistical uncertainty (one standard deviation) of the measured number of condensed atoms (horizontal error bars) and the temperature (vertical error bars).

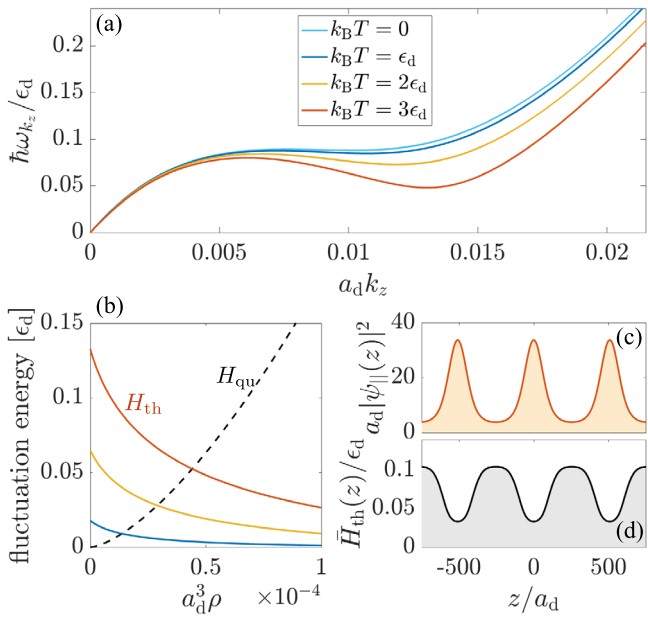

**Fig. 2 | Roton softening by thermal fluctuations.** Raising the temperature of a dipolar quantum fluid can induce a pronounced roton-maxon spectrum of its collective excitations, as shown in **a** for an infinitely elongated condensate along the z-axis [see Fig. 1c]. Heating the fluid tends to lower the energy of the roton minimum and eventually softens the roton excitation as the temperature increases. This effect can be traced back to the density dependence of the energy correction caused by fluctuations, shown in **b**. While quantum fluctuations yield an energy $H_{qu}$ (dashed line) that increases with a rising condensate density $\rho = |\psi|^2$, the contribution $H_{th}$ from thermal fluctuations decreases (solid lines). The thermal energy correction $H_{th}(\mathbf{r})$, therefore, acts as a focusing nonlinearity that supports the formation of regular density modulations. This is illustrated in **c**, **d**, where we show the axial density $\rho_{\|}(z) = \int dx dy \rho(\mathbf{r})$ along with the axial potential $\bar{H}_{th} = \rho_{\|}^{-1} \int dx dy \rho(\mathbf{r}) H_{th}(\mathbf{r})$, respectively. The calculations are performed for $a/a_d = 0.7$ and $\mu = \varepsilon_d$.

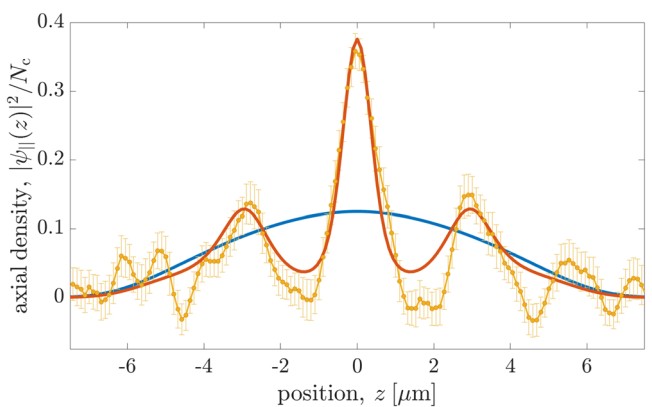

**Fig. 3 | Emergence of density modulations in trapped condensates.** Our measured axial density $|\psi_{\|}(z)|^2$ (orange points), observed for $T = 76.5$ nK and $N_c = 13,400$ condensate atoms, demonstrates the formation of a supersolid state and agrees well with the numerical simulation of the TeGPE (red line). On the other hand, equivalent simulations at zero temperature (blue line) disagree qualitatively and instead yield an unmodulated superfluid phase. The vertical error bars indicate the statistical uncertainties (one standard deviation) of the experimentally measured density.

roton energy and facilitates the formation of a density-modulated phase, as illustrated in Fig. 2c, d.

### Experimental observations in confined dipolar quantum fluids

We recently observed experimental signatures of this effect by studying the cooling–heating lifecycle of bosonic dysprosium atoms at ultralow temperatures[8]. The experiment starts from a thermal cloud of $10^5$ atoms in an optical dipole trap with trap frequencies $\omega_{x,y,z} = 2\pi \times (88, 141, 36)$ s$^{-1}$ that is elongated along the z-axis. A magnetic field is applied along the y-direction and defines the orientation of the atomic dipoles. We have traced the time evolution of the gas as it is cooled evaporatively to quantum degeneracy by lowering the depth of the trap. During the continual cooling and thermalization, we observed the expected emergence of supersolidity, and studied the equilibrium states of the quantum fluid across the supersolid phase transition. The measured density profiles indicate a higher degree of modulation at higher temperatures. While this has cast mystery on the origin of the observations, they can now be used to corroborate and benchmark our theoretical understanding. In Fig. 1b, we trace the measured contrast of the axial density modulations (see Methods Section) during the cooling process for different temperatures and condensed atom numbers. The results confirm the formation of a supersolid phase with increasing temperature in good agreement with the theoretical transition line obtained numerically by the TeGPE. Moreover, Fig. 3 compares our theoretical prediction to the measured axial density, $\rho_{\|}(z)$, for a given temperature and atom number during the cooling process[8]. The predicted zero-temperature ground state corresponds to an unstructured superfluid and deviates qualitatively from the observed supersolid state. The result of our finite-temperature TeGPE simulation, however, agrees with the experiment and reproduces quantitatively the period and amplitude of the measured density modulations. This remarkable level of agreement offers strong indication that the observed supersolid has indeed been generated by the finite temperature of the atoms.

The possibility to make detailed comparisons between theory and experiments opens up several directions for exploring the surprising thermodynamic behavior of quantum ferrofluids. Already, the ground state phase diagram exhibits a rich structure, including first-order as well as second-order quantum phase transitions in one and two-dimensional systems[29,44,45]. This offers a promising starting point for investigating how thermal fluctuations influence the nature of the fluid-solid phase transition and may affect the physics of higher dimensional supersolids[46,47], which can come in a diverse range of complex patterns[24,25,44]. Our present findings motivate future experiments to systematically explore the thermodynamic phase diagram, e.g., by actively heating the condensate across the superfluid-supersolid phase transition. Such measurements as well as first-principle simulations[48,49] would permit to expand the phase diagram of Fig. 1 into the high-temperature domain and draw a direct connection to the more familiar physics of liquid–solid phase transitions in the absence of superfluidity. Such numerical approaches may also reveal how the present phenomenology extends into the regime of strong interactions, which is becoming accessible in experiments with ultracold polar molecules[23,33,50–52]. Equally important, an improved understanding of finite-temperature effects in dipolar quantum fluids could help resolving current questions about quantitative discrepancies between measurements and theory[3,34].

## Methods

### The nonlinear wave equation

The grand canonical potential is minimal in equilibrium such that we can minimize Eq. (2) with respect to the condensate wave function $\psi(\mathbf{r})$. This yields the nonlinear wave equation

$$\mu\psi(\mathbf{r}) = \left( -\frac{\hbar^2\nabla^2}{2m} + U(\mathbf{r}) + \frac{4\pi\hbar^2 a}{m}|\psi(\mathbf{r})|^2 \right.$$
$$\left. + \int d\mathbf{r}' V_{dd}(\mathbf{r} - \mathbf{r}')|\psi(\mathbf{r}')|^2 + H_{qu}(\mathbf{r}) + H_{th}(\mathbf{r}) \right)\psi(\mathbf{r}), \tag{4}$$

which determines the equilibrium state of the condensate for a given chemical potential $\mu$. We consider a harmonic trapping potential $U(\mathbf{r}) = \frac{m}{2}(\omega_x x^2 + \omega_y y^2 + \omega_z z^2)$, with trapping frequencies $\omega_{x,y,z}$ along the three cartesian axes. The first four terms correspond to the Gross–Pitaevskii equation that describes the mean-field physics of the condensate at zero temperature. The next term is given by $H_{qu}(\mathbf{r}) = \gamma_{qu}|\psi(\mathbf{r})|^3$ and accounts for leading order effects of quantum fluctuations with a strength $\gamma_{qu}$ that increases with $a$ and $a_d$ [15,16,18] (see also Supplementary Information). Finite-temperature effects are captured by the last term as given in Eq. (3). A more detailed derivation of Eq. (4) is discussed in the Supplementary Information. We note here that the applied local-density approximation can cause an infrared divergence of the momentum integral in Eq. (3). However, the finite system size of trapped systems yields a natural momentum cutoff that ensures converged results. Indeed, we find that our calculated condensate wave functions are not sensitive to the precise choice of the momentum cutoff for relevant trap geometries (see Supplementary Information).

### Finite-temperature simulations

We have calculated the condensate wave function at finite temperatures by simulating the imaginary time evolution of the wave equation (4). More concretely, we replace $\mu\psi$ by $-\partial_t\psi$ in Eq. (4) and simulate the time evolution until $\psi(\mathbf{r},t)$ reaches a steady state for a given norm $N_c = \int d\mathbf{r}|\psi(\mathbf{r},t)|^2$. $N_c$ corresponds to the number of condensate atoms under 3D confinement as considered in Figs. 1(b) and 3, and yields the axial density $N_c/L = L^{-1}\int dx\,dy\int_0^L dz|\psi(\mathbf{r},t)|^2$ for a given length $L$ of the periodic simulation box as considered in Figs. 1a and 2. Finally, we determine the chemical potential from Eq. (4) in order to construct the thermodynamic phase diagram shown in Fig. 1a. The results shown in Figs. 1b and 3 are obtained for the experimental trap parameters $\omega_x/2\pi = 88$ Hz, $\omega_y/2\pi = 141$ Hz, and $\omega_z/2\pi = 36$ Hz, and a scattering length $a = 4.7$ nm. The simulations of Figs. 1a and 2 have been performed for $w_x = 0.0717\varepsilon_d$, $w_y = 0.142\varepsilon_d$, and $w_z = 0$. In all cases, the dipoles are considered to be polarized along the $y$-axis.

### Experimental determination of the density, temperature and atom number

We probe the atomic cloud using two different imaging techniques: (i) absorption imaging in time-of-flight measurements after releasing the atoms from the trap, and (ii) in situ measurements of the atomic density in the trap via phase-contrast imaging.

**Absorption imaging.** We turn off the optical dipole trap and probe the atomic cloud via absorption imaging after a time of flight of 26 ms. Within this expansion time the density has decreased sufficiently to perform accurate absorption measurements with resonant laser light that we shine horizontally. The recorded absorption images exhibit a characteristic bimodal profile, consisting of a broad distribution, that stems from the thermal atoms, and a narrower pattern that reflects the momentum distribution of the Bose-Einstein condensate. The broad thermal distribution permits to extract the temperature $T$ and the number $N_{th}$ of thermal atoms by fitting to a 2D Bose-enhanced Gaussian[53]. Since the total number, $N$, of atoms can be determined by integrating the total absorption signal, we can also obtain the number $N_c = N - N_{th}$ of condensed atoms, as used in Figs. 1b and 3.

**Phase-contrast imaging.** In order to measure the in-situ density profile of the trapped atoms, we use far-detuned laser light and probe the atomic density profile via Faraday phase-contrast imaging[54] with a vertically propagating probe beam (along the $y$-axis). We integrate the recorded signal along the $x$-axis to obtain an image of the 1D axial density. The position of the atomic cloud in a given image fluctuates from shot to shot. We correct for such unavoidable center-of-mass fluctuations, by realigning the central maximum of each image in the

supersolid phase to the origin, $z = 0$. By averaging over many such images, we obtain the axial density $\rho_\parallel(z)$ of the atoms, as shown in Fig. 3. Despite the large frequency detuning of the probe light, the high atomic density and high density gradients in the supersolid phase can cause lensing effects around the droplets. The resulting image distortion can generate spurious negative values of the observed optical density in the low-density regions between the droplets, as can be seen in Fig. 3. Yet, in our measurements, this distortion effect remains sufficiently small to reliable detect the transition to a modulated state and to enable direct comparisons of its characteristic length scale and modulation contrast with the theoretical predictions.

### Determination of the modulation contrast

The density modulation is quantified by the Fourier transform, $\tilde{\rho}_\parallel(k) = \int e^{-ik_z z}\rho_\parallel(z)dz$ of the axial density. We obtain the modulation contrast $\mathcal{M}$ as the ratio between the modulus of the Fourier component at the modulation wave vector and the density $|\tilde{\rho}_\parallel(0)|$ at $k_z = 0$. We apply this procedure to to our measured and numerically simulated density profiles, from which we obtain the theoretical and experimental transition shown in Fig. 1b. Hereby, the modulated states are characterized by a contrast $\mathcal{M} > 0.2$, while unmodulated condensates yield smaller values of $\mathcal{M} < 0.2$, (see also Supplementary Information).

## Data availability

The data on which the plots in this paper are based and other findings of this study are available from the authors upon request.

## Code availability

The codes on which the calculations within this paper are based and other findings of this study are available from the corresponding author, J.B., upon reasonable request.

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

## Acknowledgements
We thank Yongchang Zhang, Georg Bruun, and Jordi Boronat for valuable discussions and the Er-Dy team in Innsbruck for experimental support. This work was supported by the DNRF through the Center of Excellence for Complex Quantum Systems (CCQ) (Grant agreement no.: DNRF156), by the Carlsberg Foundation through the 'Semper Ardens' Research Project QCooL, and by the DFG through the SPP1929 GiRyd. F.M. acknowledge support through the grant PID2021-128910NB-I00 of the Ministerio de Ciencia e Innovación. J.S.-B. acknowledges funding by the European Union, by the Spanish Ministry of Universities and by the Recovery, Transformation and Resilience Plan through a grant from Universitat Politècnica de Catalunya. F.F. and C.P. acknowledge support through an ERC Consolidator Grant (RARE, No. 681432), the QuantERA grant MAQS (No. I4391-N), and the FOR grant (2247/PI2790) by the Austrian Science Fund FWF.

## Author contributions
This project was conceived by F.F. and T.P. The theoretical aspects of this work were developed by J.B. and F.M.; J.B. conducted the numerical calculations. The experimental analysis was performed by C.P. All authors contributed to the technical discussions and the writing of the manuscript.

## Competing interests
The authors declare no competing interests.
