## [Peer Review File · Nature Communications]

Heating a dipolar quantum fluid into a solidREVIEWER COMMENTS

Reviewer #1 (Remarks to the Author):

This is an interesting and timely manuscript of finite-temperature effects in dipolar superfluids. Such systems of ultracold quantum systems are being studied in detail, due to their combination of interesting properties, such as the emergence of both superfluidity and supersolidity, which makes this a topic of rather broad interest. The present work uses appropriate methodology to make significant advances to our understanding of the role of finite temperature on the phase diagram of such systems: specifically this work reports (theoretically) on the finite-temperature phase diagram of such systems, which, as temperature is increased, reveals the emergence of supersolidity (rather than superfluidity) for smaller condensate atom numbers. This is then shown to be consistent with experimental measurements, which acts as further validation of the obtained results, and helps shed more light into previous experimental observations. This work is well written, accessible to a broad audience, and sufficiently novel and timely to potentially merit, in some form, publication in Nature Communications.

Below, I address some issues that the authors should further comment upon:

1. Can the authors clarify if the experimental data points shown in Fig. 1(b) [and Fig. 3] are old experimental measurements (e.g. taken from Ref. [41]) which have been re-analysed/presented in a manner facilitating direct comparison with presented numerical findings? This is my current understanding – but, assuming so, this is not very clearly stated neither in the main manuscript, nor in the Methods section. Assuming that to be so, please make such connections more explicit. If that is not the case, i.e. if this is newly-obtained data inspired by previous experiments/existing apparatus, then this should also be clarified. In principle, in such work I would have preferred (and partly expected) to see more direct experimental evidence for the postulated transition, i.e. by means of increasing T at (nearly) fixed N_c . Assuming reported data to be existing ones, I understand why the authors have not done this here, and assume the authors will aim to perform such exciting experimental study in the near future.

2. Schematic in Fig. 1(c) is indeed very nice and relevant, but perhaps the authors could clarify to which (if any) points in Fig. 1(b) this corresponds to?

3. In the Supplementary Info (which is very useful), I would have liked to have seen more evidence / a better discussion in the context of Fig. S2: For example (i) why is Fig. S2(a) presented in an “inverted” x-axis manner to Fig. 1(a); (ii) better explanations should be given about what is plotted in Fig. S2(a), i.e. what the different colours represent, etc.; (iii) I would have expected to see both (a)-(b) plots shown in same axes as in main paper, and (iv) Fig. S2(b) compared more clearly to Fig. 3 [e.g. shaded area in S2(b)]

does not correspond to shaded area in Fig. 3, but rather to the (highly relevant) numerical calculations – but, on first inspection, such presentation is rather confusing.]

4.A minor point: Ref. [2] in SI is incomplete.

Reviewer #2 (Remarks to the Author):

The manuscript stems from an international theory-experiment collaboration. It describes unexpectedly how heating a dipolar superfluid from near-zero temperatures can induce a phase transition to a supersolid state with a broken translational symmetry. The manuscript presents new results on a topic, which is of general interest for the public. Furthermore, Fig. 1(b) and Fig. 3 present an impressive comparison between experimental measurements and theoretical modelling. For all these reasons a publication of the manuscript in Nature Communications is justified.

However, there is one point which needs clarification. The main new feature of the manuscript is the discussion of the impact of thermal fluctuations upon the formation of a supersolid. And to this end a precise measurement of the temperature in the laboratory is an indispensable prerequisite:

1) Do the experimental data shown in Fig. 1(b) stem originally from Ref. [41]? If the answer is yes, then the caption of Fig. 1(b) should mention that explicitly.

2) According to my knowledge Ref. [41] and this manuscript are the first ones to report about temperature measurements for supersolids. Therefore, it should be indicated briefly how the temperature was measured. Maybe the authors could indicate the difficulty in previous temperature measurements.

Reviewer #3 (Remarks to the Author):

In "Heating a quantum fluid into a solid" J. Sánchez-Baena and the co-authors investigate the superfluid-supersolid phase diagram of ultracold dipolar (dysprosium) atoms. They find that contrary to intuition an increase in temperature can lead to a superfluid-to-supersolid phase transition. They can explain this behavior with thermal fluctuations of the system. The work thus presents a theoretical framework to understand the unexpected experimental results of the earlier work ref. [41]. The results are noteworthy as they advance our understanding on the impact of microscopic fluctuations on the macroscopic state of matter. They take the established knowledge on quantum fluctuations of

condensates and expand it by additionally including thermal fluctuations, the necessary step for the study of finite-temperature systems.

The manuscript is generally well written, easy to read and of refreshing compactness. There are only a few places where some modifications to the text would increase the clarity of the presentation and one point concerning the data analysis which I would like to ask the authors to address.

1. page 1, first paragraph: In the sentence “This finding sheds light on recent experimental observations” the appropriate references to the intended experiments should be provided.

2. page 2, equation (1): A curiosity, what is the value of θ for the experiment at hand? How does it affect the results?

3. Figure 1 (b): This Figure represents the central result of the present work. Unfortunately, it is in my opinion also the most “unfortunate” of them all.

3a. First I do not understand why the vertical axis is located on the right side. At first I thought that it would be an alternative scale to the vertical axis of part (a), but I think that is not true. So placing the axis on the left would avoid this ambiguity.

3b. My second concern is the blue-orange modulation color scale. I understand that the color transition point is probably chosen to support the analysis result but having absolutely no color variation in the range, say, 0.2-0.6 beats the purpose of presenting a color scale in the first place. I ask the authors to choose a color scale that allows for a real estimation of the measured modulation strength. (There are ample bi-color scales that still have some variation of the color in each sector.)

3c. Generally, the data points are those of Fig. 4 in ref. [41]. This should be made clearer as for me I only really understood the present Figure after seeing the “original” data. It was also beneficial to appreciate the observed phase transition and to understand the choice of the color scale. Ideally, this data (modulation vs. N_c) would be reproduced in the present manuscript.

3d. It is not clear to me how to especially translate between horizontal axes of parts (a) and (b). How does N_c relate to a_d/a ? The point I am probably asking: Which part of the phase space presented in (a) is probed in (b)? It would be really nice if the corresponding area could be indicated in (a) even though I feel that we are probably in quite a different parameter range there.

4. page 3, bottom of left column: Considering the more general readership of Nature Communications, a few more words on how exactly the “thermal softening” (that is a shift of the minimum position towards larger $a_d k_z$, correct?) of the roton mode drives the “instability of the superfluid” would be appreciated.

4. page 3, first paragraph of right column: Is “focusing” nonlinearity an established expression in the field? To me it felt rather fancy and required some extra thinking to understand that basically an “attractive” force was intended. Why so complicated, or am I missing some additional point?

5. page 5, last sentence: This is my only concern of the present work that goes beyond cosmetics but concerns the experimental method. The authors state that due to technical reasons systematic errors in the measured densities occur. That is perfectly fine. What is not so fine is to just drop those data where the densities are negative (and which would clearly appear non-physical in the paper). However, it is not that only this data is affected by the technical limitations. I expect all data to deviate from the actual density to some degree, just that there it is less apparent. So just dropping obviously wrong data and leaving the rest as-is does not seem appropriate to me. Instead, one should try to estimate the error in the data and present all the data together with its error. I understand that this easily leads to a mess in the representation of Fig. 3, but I believe that this is a challenge that should be accepted.

To conclude, the authors present a very nice work that explains previously puzzling experimental results and that widens our understanding of fluctuations in general and dipolar quantum fluids in particular. As such I am, after the authors could address my concerns stated above, in favor of publication of the manuscript in Nature Communications.

Response to Reviewer #1

We would like to thank the reviewer for the detailed assessment of our manuscripts and are very happy about the positive feedback on our results. We also thank the reviewer for the helpful remarks, which we have all addressed as detailed below.

1. Can the authors clarify if the experimental data points shown in Fig. 1(b) [and Fig. 3] are old experimental measurements (e.g. taken from Ref. [41]) which have been re-analysed/presented in a manner facilitating direct comparison with presented numerical findings? This is my current understanding – but, assuming so, this is not very clearly stated neither in the main manuscript, nor in the Methods section. Assuming that to be so, please make such connections more explicit. If that is not the case, i.e. if this is newly-obtained data inspired by previous experiments/existing apparatus, then this should also be clarified. In principle, in such work I would have preferred (and partly expected) to see more direct experimental evidence for the postulated transition, i.e. by means of increasing T at (nearly) fixed N_c . Assuming reported data to be existing ones, I understand why the authors have not done this here, and assume the authors will aim to perform such exciting experimental study in the near future.

As the reviewer describes correctly the experimental data has been obtained from the experiment reported in Ref. [41] (now Ref.[8] in the revised manuscript). In order to compare to our theory and draw conclusions about the predicted transition we have reanalyzed the measurement and obtained the data presented in the main text and in the supplementary material. For example, this includes the axial density shown in Fig.3, which has not been presented in Ref.[8], and now been obtained from the data of the experiment reported in [8].

We thank the reviewer for pointing out that this can be stated more clearly and have revised the manuscript accordingly. Specifically, we now cite the corresponding reference much earlier, already in the first paragraph, and cite Ref.[8] in the caption of Fig1 when discussing the presented data. While we have already tried refer to Ref.[8] when discussing the data throughout the manuscript, we have added further reference and additional text to the paragraph describing the theory-experiment comparisons in Fig.1 and Fig.3.

We agree with the reviewer that temperature scans at fixed atom numbers would be an exciting measurement to trace the predicted transition more accurately. Currently, the variation of the temperature and atom number results from the evaporative cooling process and cannot be controlled

independently, which makes it inherently difficult to scan the phase diagram in a different manner. However, we completely agree that implementing different approaches to vary the temperature at constant atom number would be an exciting experiment that should be pursued in the future. We thank the reviewer for pointing this out and now mention this excellent point in the concluding paragraph.

2. Schematic in Fig. 1(c) is indeed very nice and relevant, but perhaps the authors could clarify to which (if any) points in Fig. 1(b) this corresponds to?

The drawing of Fig.1c is only meant to illustrate the underlying process schematically and does not correspond to actual data of panels (a) or (b). This is now stated more explicitly in the figure caption.

3. In the Supplementary Info (which is very useful), I would have liked to have seen more evidence / a better discussion in the context of Fig. S2: For example (i) why is Fig. S2(a) presented in an “inverted” x-axis manner to Fig. 1(a); (ii) better explanations should be given about what is plotted in Fig. S2(a), i.e. what the different colours represent , etc.; (iii) I would have expected to see both (a)-(b) plots shown in same axes as in main paper, and (iv) Fig. S2(b) compared more clearly to Fig. 3 [e.g. shaded area in S2(b) does not correspond to shaded area in Fig. 3, but rather to the (highly relevant) numerical calculations – but, on first inspection, such presentation is rather confusing.]

- (i): That is an excellent point and we have now changed the axis in Fig.S2 to match the one in Fig. 1.
- (ii): We have modified the caption of Fig.S2 to better explain the colored areas in Fig. S2(a)
- (iii): We now use the same axis limits in Fig.S2 as we use in Fig.1 and Fig.3 of the main text.
- (iv): We have modified the figure accordingly. We have removed the colored shading and now use different line styles for the different calculation results.

4. A minor point: Ref. [2] in SI is incomplete.

We thank the reviewer for spotting this mistake and have corrected the reference.

Response to Reviewer #2

We would like to thank the reviewer for the thorough assessment of our manuscript. We are very happy about the reviewer's positive outlook on our work, finding that publication in Nature Communications is justified. The reviewer has raised two very good points, which we have both addressed as detailed below.

1. Do the experimental data shown in Fig. 1(b) stem originally from Ref. [41]? If the answer is yes, then the caption of Fig. 1(b) should mention that explicitly.

We thank the reviewer for pointing out that this should be stated more clearly and have modified the caption of Fig. 1(b) accordingly. In addition, we now cite the corresponding reference much earlier, already in the first paragraph, and have added further reference and additional text to the paragraph describing the theory-experiment comparisons in Fig.1 and Fig.3.

2. According to my knowledge Ref. [41] and this manuscript are the first ones to report about temperature measurements for supersolids. Therefore, it should be indicated briefly how the temperature was measured. Maybe the authors could indicate the difficulty in previous temperature measurements.

We thank the reviewer for the suggestion and have now added a description of the temperature measurements in the Methods Section. In brief, to extract T and N_{th} , we record for each set of parameters (i.e., for each point in Fig.1b), an absorption image of the expanded atomic cloud using horizontal imaging. This is the standard method in quantum-gas experiments to extract T and N_{th} . The atomic distribution in the absorption images is bimodal with a dense modulated central peak corresponding to the condensed atoms in the supersolid and a broad contribution given by the thermal component. The latter can be fitted using a 2D Bose-enhanced Gaussian function. From this fit, we estimate the temperature of the cloud and the number of thermal atoms N_{th} .

Response to Reviewer #3

We would like to thank the reviewer for the careful assessment of our manuscript and are happy about the reviewer's feedback and conclusion about publication in Nature Communication. Also, we very much appreciate the detailed and constructive remarks by the reviewer. As detailed below, we have addressed all of the reviewer's points and revised the manuscript accordingly.

1. Page 1, first paragraph: In the sentence "This finding sheds light on recent experimental observations" the appropriate references to the intended experiments should be provided.

We thank the reviewer for pointing this out and have added the respective reference.

2. page 2, equation (1): A curiosity, what is the value of θ for the experiment at hand? How does it affect the results?

The dipoles are aligned perpendicular to the axis of the elongated trap geometry considered in the theory and realized in the experiment. More specifically, the dipoles are aligned along the y-axis, which then defines the angle θ for a given three-dimensional distance vector \mathbf{r} between two atoms. In the experiment, the dipole orientation is set by the external magnetic field, which is aligned along the y-direction. The atomic cloud in the cigar-shaped trap is elongated along the z-axis. We agree with the reviewer that this offers useful information and have now added additional text to specify the experimental trap geometry and considered dipole orientation. The orientation of the dipoles along the y-axis is also mentioned in the second paragraph of the Methods section.

3a. First I do not understand why the vertical axis is located on the right side. At first I thought that it would be an alternative scale to the vertical axis of part (a), but I think that is not true. So placing the axis on the left would avoid this ambiguity.

We have moved the axis label to the left vertical axis in Fig.1b.

3b. My second concern is the blue-orange modulation color scale. I understand that the color transition point is probably chosen to support the analysis result but having absolutely no color variation in the range, say, 0.2-0.6 beats the purpose of presenting a color scale in the first place. I ask the authors to choose a color scale that allows for a real estimation of the measured modulation strength. (There are ample bi-color scales that still have some variation of the color in each sector.)

The (somewhat unusual) color scale has been chosen to focus on the transition between the unmodulated and modulated state. As the reviewer points out correctly, this necessarily suppresses information about the detailed values of the modulation contrast and we certainly agree that this would present interesting additional information. We have therefore removed the color scale in Fig.1b and now show unmodulated states by blue points and modulated states by orange points. In fact, this matches the procedure to obtain the theoretical transition line and, thus, also simplifies the comparison between theory and experiment.

In order to show the explicit values of the modulation, we have now added a figure and corresponding discussion to the supplementary material, where we compare the experimental values and theoretical prediction for the modulation contrast.

We thank the reviewer for raising this excellent point and hope that the corresponding revisions have improved the presentation of the results.

3c. Generally, the data points are those of Fig. 4 in ref. [41]. This should be made clearer as for me I only really understood the present Figure after seeing the “original” data. It was also beneficial to appreciate the observed phase transition and to understand the choice of the color scale. Ideally, this data (modulation vs. N_c) would be reproduced in the present manuscript.

Following the reviewer’s suggestion, we have revised the caption of Fig.1 and several places of the main text to state this more clearly. Specifically, we now cite the corresponding reference much earlier, already in the first paragraph, and cite Ref.[8] in the caption of Fig1 when discussing the presented data. While we have already tried refer to Ref.[8] when discussing the data throughout the manuscript, we have added further reference and additional text to the paragraph describing the theory-experiment comparisons in Fig.1 and Fig.3.

Moreover, the mentioned data is now presented in the supplementary information where we show the modulation contrast in theory and experiment (see point 3) above), which we hope now brings more clarity.

3d. It is not clear to me how to especially translate between horizontal axes of parts (a) and (b). How does N_c relate to a_d/a ? The point I am probably asking: Which part of the phase space presented in (a) is probed in (b)? It would be really nice if the corresponding area could be indicated in (a) even though I feel that we are probably in quite a different parameter range there.

The two horizontal axes cannot be directly related to each other. In panel (a), we consider an infinitely elongated condensate and keep the chemical potential fixed, while varying a_d/a . Panel (b) is for a finite-sized atomic cloud where a_d/a is kept fixed while varying the number of condensed atoms. It can be noted that the experimental value of $a_d/a = 1.46$ (now given explicitly in the caption) in panel (b) is covered in the phase diagram of (a), such that both panels cover similar parameter regions. Generally, changing a_d/a as well as changing N_c , both changes the potential energy contribution of the dipole-dipole interaction and therefore has a similar effect on the phase diagram as seen in the figure. However, they are independent parameters and cannot be related to each other through simple scaling. Most importantly, though, both panels show that a modulated state emerges upon raising the temperature.

4a. Page 3, bottom of left column: Considering the more general readership of Nature Communications, a few more words on how exactly the “thermal softening” (that is a shift of the minimum position towards larger $a_d k_z$, correct?) of the roton mode drives the “instability of the superfluid” would be appreciated.

Here, mode softening refers to the lowering of the minimum energy, i.e. the roton energy. As this energy minimum decreases with increasing temperature, the modulated state (corresponding to the roton momentum at the energy minimum) becomes energetically more favorable, which eventually leads to the observed phase transition. We thank the reviewer for pointing out that such further discussions would be useful and have added corresponding text to the mentioned paragraph in the revised manuscript.

4b. Page 3, first paragraph of right column: Is “focusing” nonlinearity an established expression in the field? To me it felt rather fancy and required some extra thinking to understand that basically an “attractive” force was intended. Why so complicated, or am I missing some additional point?

The term “focusing nonlinearity” is commonly used in the description of nonlinear wave dynamics. The reviewer is correct that a focusing nonlinearity can arise from an attractive force and would generally act similarly to an attractive interaction. However, there is no exact equivalence since the precise dependence of H_{th} on density is not identical to the meanfield energy of attractively interacting particles. For this reason, we used the more general term of a “focusing nonlinearity”.

Following the reviewer’s suggestion, we have revised the corresponding sentence to simplify the formulation.

5. Page 5, last sentence: *This is my only concern of the present work that goes beyond cosmetics but concerns the experimental method. The authors state that due to technical reasons systematic errors in the measured densities occur. That is perfectly fine. What is not so fine is to just drop those data where the densities are negative (and which would clearly appear non-physical in the paper). However, it is not that only this data is affected by the technical limitations. I expect all data to deviate from the actual density to some degree, just that there it is less apparent. So just dropping obviously wrong data and leaving the rest as-is does not seem appropriate to me. Instead, one should try to estimate the error in the data and present all the data together with its error. I understand that this easily leads to a mess in the representation of Fig. 3, but I believe that this is a challenge that should be accepted.*

We thank the reviewer for raising this question and pointing out that more details and explanation would be helpful. Following the reviewer’s remarks, we now provide such addition information in the revised manuscript.

The appearance of a seemingly negative optical density is common to in-situ imaging measurements and typically occurs for small atomic densities near high-density regions. Correspondingly, in our case, such spurious negative densities can occur in between droplets of the supersolid state. There are different technical reasons that can cause negative densities in an individual image, including fluctuations of the background light, small misalignment of the high-resolution objective (leading to aberration or interference), and, most notably, lensing effects.

We probe the in-situ density by illuminating the atomic cloud with far-detuned laser light and performing phase contrast imaging. Hereby, the laser detuning is chosen such that optical absorption is greatly suppressed while the phase shift due to the index of refraction remains significant and can be used to measure the density profile (see e.g. Ref. [54]). Therefore, the real part of the refractive index

can lead to small but finite lensing effects in the presence of large density gradients as is the case in the supersolid phase. The resulting refraction of the probe beam can lead to a slight distortion of the images, that leaves the probing of the high-density regions largely unaffected but can cause spurious negative values of the measured density in the low-density regions between the droplets. This is a commonly encountered effect, but difficult to correct without accurate prior knowledge of the actual density profile. As in previous works, we have therefore chosen to set negative density values in single images to zero, which yields an average density profile as was shown in Fig.3 of the previously submitted manuscript.

We thank the reviewer for the above remark and fully agree that displaying the bare data, without correcting for negative-density errors, would offer a more direct comparison. Therefore, we followed the reviewer's advice and now show this data including technical imperfections along with statistical errors in Fig.3. We have also added a dedicated part about the optical probing (in-situ and time-of-flight) of the BEC in the Methods Section.

REVIEWERS' COMMENTS

Reviewer #1 (Remarks to the Author):

The authors have appropriately clarified my previous queries, and responded satisfactorily to similar/further queries by the other reviewers.

In my opinion, the revised manuscript can be published as is.

Reviewer #3 (Remarks to the Author):

In the revised manuscript "Heating a quantum fluid into a solid" the authors quite thoroughly addressed the concerns I raised in my first review of their work. In particular I am very happy about the improved Fig. 1 in conjunction with the new Fig. S3, and the improved discussion of the uncertainties of Fig. 3. Together with the other improvements throughout the manuscript that should enhance the accessibility of the work, I am now very much in favor of recommending the manuscript for swift publication in Nature Communications.

We thank all reviewers for the positive feedback on our work and for recommending publication without additional requests and comments.